# SAFER REINFORCEMENT LEARNING WITH COUNTEREXAMPLE-GUIDED OFFLINE TRAINING

## ABSTRACT

Safe reinforcement learning (RL) aims at addressing the limitation of reinforcement learning in safety-critical scenarios, where failures during learning may incur high costs. Several methods exist to incorporate external knowledge or to use proximal sensor data to limit the exploration of unsafe states. However, dealing with (partially) unknown environments and dynamics, where an agent must discover safety threats during exploration, remains challenging. In this paper, we propose a method to abstract hybrid continuous-discrete systems into compact surrogate models representing the safety-relevant knowledge acquired by the agent at any time during exploration. We exploit probabilistic counterexamples generation to synthesise minimal, partial simulation environments from the surrogate model where the agent can train offline to produce heuristic strategies to minimise the risk of visiting unsafe states during subsequent online exploration. We demonstrate our method's effectiveness in increasing the agent's exploration safety on a selection of problems from literature and the OpenAI Gym.

## 1 INTRODUCTION

A critical limitation of applying Reinforcement Learning (RL) in real-world control systems is its lack of a guarantee to avoid unsafe behaviours. At its core, RL is a trial-and-error process, where the learning agent explores the decision space and receives rewards for the outcome of its decisions. However, in safety-critical scenarios, failing trials may result in high costs or unsafe situations and should be avoided as much as possible. Nevertheless, several learning methods try to incorporate the advantages of model-driven and data-driven methods to encourage safety during learning (Kim et al., 2020; García & Fernández, 2015).

One natural approach for encouraging safer learning is to analyse the kinematic model of the learning system with specific safety requirements and to design safe exploration (Garcia & Fernández, 2012; Alshiekh et al., 2018; Pham et al., 2018) or safe optimisation (Achiam et al., 2017; Tessler et al., 2018; Stooke et al., 2020) strategies that avoid unsafe states or minimise the expected occurrence of unsafe events during training. However, this approach is not applicable for most control systems with partially-known or unknown dynamics, where not enough information is available to characterise unsafe states or events a priori.

To increase the safety of learning in environments with (partially) unknown dynamics, we propose an online-offline learning scheme where online execution traces are used to construct an abstract representation of the visited state-space and the outcome of the agent's actions. The abstraction is iteratively refined with evidence collected online and constitutes a compact stochastic model of safety-relevant aspects of the environment. A probabilistic model checker is then used to produce from the abstract representation a minimal counterexample sub-model, i.e., a subset of the abstract state space and of the related action space within which the agent can reach unsafe states with a probability larger than tolerable. Simulation-based learning is then used to train the agent offline within diverse counterexample sub-models to refine its strategy towards avoiding unsafe states. This way, we migrate most trial-and-error risks to the simulation environment, while discouraging the exploration of risky behaviours during online learning.

Our main contribution in this paper is a safer RL method for control systems with no prior knowledge of the environment relying on probabilistic counterexample guidance. In particular, we 1) propose a conservative geometric abstraction model representing safety-relevant experience collected by the agent at any time during online exploration, with theoretical convergence and accuracy guarantees, 2) use minimal label set probabilistic counterexample generation to synthesise small-scale simulation environments for the offline training of the agent aimed at reducing the likelihood of reaching

unsafe states during online exploration, 3) a preliminary evaluation of our method to enhance a Q-Learning agent on problems from literature and the OpenAI Gym demonstrating how it achieves comparable cumulative rewards while increasing the exploration safety rate by up to 40% compared with previous related work (Hasanbeig et al., 2018; 2022).

## 2 Background

**Markov Decision Process.** We represent the abstract model of a control system and its environment, with hybrid discrete-continuous state space, as a (discrete) MDP (Bellman, 1957): a tuple $< S, s_0, A, P, R, L >$, where $S$ is a finite state space, $s_0$ is the initial state, $A$ is a finite set of discrete actions, $P : S \times A \times S \to [0, 1]$ is the probability of transitioning from a state $s \in S$ to $s' \in S$ with action $a \in A$, $R : S \times A \to \mathbb{R}$ is an immediate reward function, $L : S \to 2^{AP}$ is a labelling function that assigns atomic propositions (*AP*) to each state.

A policy $\pi : S \to A$ selects in every state $s$ the action $a$ to be taken by the agent. Given a discount factor $\gamma \in (0, 1]$, such that rewards received after $n$ transitions are discounted by the factor $\gamma^n$, a deterministic policy selects in state $s \in S$ the action $\arg\max_a Q^\pi(s, a)$ that maximises the Q-value (Watkins & Dayan, 1992) defined as:

$$Q^\pi(s, a) = r(s, a) + \gamma \mathbb{E}_{s_{t+1} \sim P(s_{t+1}|s,a), a_{t+1} \sim \pi(a_{t+1}|s_{t+1})}[Q^\pi(s_{t+1}, a_{t+1})] \tag{1}$$

**Probabilistic Model Checking.** Probabilistic model checking is an automated verification method that, given a stochastic model – the MDP in our case – and a property expressed in a suitable probabilistic temporal logic, can verify whether the model complies with the property or not (Baier & Katoen, 2008). In this work, we use Probabilistic Computational Temporal Logic (PCTL) (Hansson & Jonsson, 1994) to specify the probabilistic safety requirement for the agent. The syntax of PCTL formulae is recursively defined as:

$$\Phi := true \mid \alpha \mid \Phi \wedge \Phi \mid \neg\Phi \mid P_{\bowtie p}\varphi \tag{2}$$

$$\varphi := X\Phi \mid \Phi U\Phi \tag{3}$$

A PCTL property is defined by a state formulae $\Phi$, whose satisfaction can be determined in each state of the model. $true$ is satisfied in every state, $\alpha \in AP$ is satisfied in any state whose label includes $\alpha \in L(s)$, and $\wedge$ and $\neg$ are the Boolean conjunction and negation operators. The modal operator $P_{\bowtie p}\varphi$ holds in a state $s$ if the cumulative probability of all the paths originating in $s$ and satisfying the path formula $\varphi$ does not exceed $p \in [0, 1]$ *under any possible policy*. The Next operator $X\Phi$ is satisfied by any path originating in $s$ such that the next state satisfies $\Phi$. The Until operator $\Phi_1 U\Phi_2$ is satisfied by any path originating in $s$ such that a state $s'$ satisfying $\Phi_2$ is eventually encountered along the path, and all the states between $s$ and $s'$ (if any) satisfy $\Phi_1$. The formula $true \ U \ \Phi$ is commonly abbreviated as $F\Phi$ and satisfied by any path that eventually reaches a state satisfying $\Phi$. A model $M$ satisfies a PCTL property $\Phi$ if $\Phi$ holds in the initial state $s_0$.

PCTL allows specifying a variety of safety requirements. In this work, we focus on safety requirements specified as upper-bounds on the probability of eventually reaching a state labelled `unsafe`:

**Definition 2.1** *Safety Specification: Given a threshold $\lambda \in (0, 1]$, the safety requirement for a learning agent is formalised by the PCTL property $P_{\leq\lambda}$ [$F$ `unsafe`], i.e., the maximum probability of reaching a state labelled as `unsafe` must be lower than or equal to $\lambda$.*

**Counterexamples in Probabilistic Model Checking.** When a model $M$ does not satisfy a PCTL property, a counterexample can be computed as evidence of the violation. In this work, we adapt the *minimal critical label set* counterexample generation method of (Wimmer et al., 2013), which computes the minimal submodel – a subset of the model state space and of the actions possible in such subset – within which there is a policy that already violates the property:

**Definition 2.2** *Counterexample of a Safety Specification: The solution of the following optimisation problem (adapted from (Wimmer et al., 2013)) is a minimal counterexample of the safety specification $P_{\leq\lambda}$ [$F$ `unsafe`]:*

$$\text{minimise} \ -\frac{1}{2}\omega_0 p_{s_0} + \sum_{\ell \in L_c} \omega(\ell)x_\ell, \ , \text{such that}$$

$$p_{s_0} > \lambda \tag{4}$$

$$\forall s \in T. \ p_s = 1 \tag{5}$$

$$\forall s \in S \setminus T. \ \sum_{a \in P(s)} \pi_{s,a} \leq 1 \tag{6}$$

$$\forall s \in S \setminus T. \ p_s \leq \sum_{a \in P(s)} \pi_{s,a} \tag{7}$$

$$\forall s \in S \setminus T, \ \forall a \in A, \ \forall \ell \in L_c(s,a,s'). \ p_{s,a,s'} \leq x_\ell \tag{8}$$

$$\forall s \in S \setminus T, \ \forall a \in A, \ p_{s,a,s'} \leq P(s,a,s') \cdot p_{s'} \tag{9}$$

$$\forall s \in S \setminus T, \ \forall a \in A. \ p_s \leq (1 - \pi_{s,a}) + \sum_{s':P(s,a,s')>0} p_{s,a,s'} \tag{10}$$

$$L_c(s,a,s') \implies P(s,a,s') > 0 \tag{11}$$

$$\forall s,s' \in S, a \in a. \ p_s \in [0,1] \land p_{s,a,s'} \in [0,1] \land \pi_{s,a} \in \{0,1\} \tag{12}$$

$$\forall \ell \in L_c(s,a,s'). \ x_\ell \in \{0,1\} \tag{13}$$

*where $S$ is the state space of the model, $T$ is the set of states labelled as* unsafe, *$p_s$ represents the probability of reaching reaching any state in $T$ from state $s$, $\pi_{s,a} \in \{0,1\}$ indicates that action $a$ is selected in state $s$, $p_{s,a,s'}$ represents the probability contribution of the transitions from $s$ to $s'$ via action $a$, where this transition selected to be part of the counterexample, $\ell \in L_c(s,a,s')$ is the label identifying a transition from $s$ to $s'$ via $a$ (not to be confused with the function $L$ labeling states of the model) such that $x_l = 1$ iff the transition is included in the counterexample, $x_l = 0$ otherwise.*

Intuitively, the optimisation problem aims to find a policy $\pi$ that will make the agent violate the safety specification within the smallest counterexample sub-model, i.e., the sub-model composed of all the transitions $(s,a,s')$ from the original model whose corresponding $x_l$ is 1.

To this goal, equation 4 requires the probability of reaching an unsafe state to be $> \lambda$ (violation of the safety specification); equation 5 fixes $p_s = 1$ for all the unsafe states; equation 6 impose the agent to select at most one action $a$ for each state $s$; if no action is chosen in a state $s$, equation 7 ensures that $p_s = 0$. equation 8 ensures that the contribution of the transition $(s,a,s')$ is 0 if such transition is not included in the counterexample, otherwise, equation 9, $p_{s,a,s'}$ is bounded by the probability of the corresponding transitions in the model ($P(s,a,s')$) times the probability of reaching the target from $s'$ ($p_{s'}$). equation 10 ensures that if action $a$ is selected in state $s$, the probability of reaching $T$ from $s$ is bounded by the sum of the probabilities of reaching $T$ from its successors given $a$. Finally, two additional constraints are defined in (Wimmer et al., 2013) to prevent the agent from getting stuck in an infinite loop that would prevent it from reaching $T$. We omit them in this section for simplicity and refer the reader to (Wimmer et al., 2013) (they are implemented for our experiments).

In the objective function, the weights $\omega(\ell)$ assign to the selector variable $x_\ell$ the opposite normalised Q-value corresponding to the state-action pair in $\ell$, while $\omega_0 > \max\{\omega(\ell)) \mid \forall \ell \in L_c \land \omega(\ell) > 0\}$. With this weighting, we encourage the selection with a preference for labels with larger Q-Value, to discover violating behaviours in the most commonly-selected state-action pairs, while also minimising the size of the sub-model.

**Multiple Counterexamples.** The minimal solution to the MILP problem in Def. 2.2 may not be unique, and different minimal solutions may lead to different violating behaviours (Češka et al., 2019). Because counterexamples will be used to generate simulation environments for offline learning, we aim to keep their size small for faster training. On the other hand, representing diverse violating behaviours may lead to more robust safety training. To foster the diversity of violating behaviour information used for guiding safer exploration, we further generate a set of "mutations" of a counterexample by adding blocking clauses to labels selected in existing counterexamples, i.e., adding constraints to the optimisation problem that force arbitrary labels selected in the initial counterexample to be deselected by forcing the corresponding $x_\ell = 0$. The goal is to produce multiple diverse counterexamples that jointly provide a more comprehensive representation of the violations scenarios so far explored by the agent.

# 3 Simulation-based Learning with Counterexample Exploration Guidance

We present our learning framework for simulation-based learning with counterexample exploration guidance in this section. The key idea of our method is to mitigate the risk of visiting discovered unsafe states by alternating online and offline training phases. The online phases aim at exploring the environment and learning the outcome of different actions to build and continuously refine an abstract, compact representation of the control problem. The offline phases use counterexamples from the abstract representation to synthesise smaller simulation environments that allow the agent to refine its policies towards minimising the risk of visiting discovered unsafe states. We first describe our abstraction procedure and the synthesis of simulation models in Sec. 3.1. Then, in Sec. 3.2 we introduce the simulation-based learning algorithm, combining offline learning with online exploration, and discuss the main challenges of offline learning.

## 3.1 Simulation Model Construction

We present the process of the simulation model construction in three parts: 1) partition different safety behaviours of a continuous state space using a geometric abstraction method based on the sampled dataset of explored paths, and 2) organise the partition results in an MDP surrogate model, and 3) construct the simulation environments for offline learning based on the surrogate model.

**Geometric Abstraction.** We assume the state to be represented by a vector of real numbers. Sampled states not recognised and labelled as `unsafe` by the agent are assumed safe to explore, and we also assume the label set of a state to expand monotonically: in particular, a state labelled as `unsafe` will be considered unsafe for the rest of the exploration. We aim at abstracting the explored concrete state space as a finite set of disjoint polyhedra, each defined as the intersection of a finite set of hyperplanes. This process can be seen as a classification task, where regions of the continuous concrete state space are covered by the union of polyhedra, with the goal of separating safe and unsafe concrete regions. While such separation can achieve arbitrary accuracy by increasing the number of polyhedra, in general, not all shapes of the concrete regions can be exactly covered, and very high accuracy may require a prohibitive computational cost. To control the impact of the limited abstraction accuracy on the safety of the agent, we require the abstraction to conservatively satisfy the following safety invariant at any time.

**Definition 3.1** *Safety Invariant. All explored unsafe concrete states should be mapped to unsafe abstract states.*

Inspired by the idea of casting the learning of geometric concepts as a set cover problem in (Bshouty et al., 1998; Sharma et al., 2013), we frame the separation task as a minimal red-blue set cover problem maintaining the safety invariant. Let $X = \{\boldsymbol{x}_0, \boldsymbol{x}_1, ..., \boldsymbol{x}_n\}$ be the set of all explored concrete states and $U = \{\boldsymbol{u}_0, \boldsymbol{u}_1, ..., \boldsymbol{u}_m\}$ be the set of explored states assigned the "unsafe" label. Find the minimum number of sets $C_i$, with false classification rate for safe concrete states $(X \setminus U)$ bounded by a constant $f$, such that $C = \{\cup_{i=1} C_i\}$, where each element $u \in U$ is covered by one set $C_i$. An element $\boldsymbol{x}$ is covered by a (polyhedra) predicate $C_i \in C$ if $\boldsymbol{x} \in C_i$, where

$$C_i = \{\boldsymbol{x} \mid \boldsymbol{\omega}\boldsymbol{x} + \boldsymbol{b} \leq 0\} \tag{14}$$

We fix $\omega = 1$ in this work, i.e., restrict to hyper-boxes rather than arbitrary polyhedra, to simplify the implementation of a branch and bound abstraction procedure. The solution set guarantees that all the unsafe concrete states are covered by a Boolean combination of $C$, while safe concrete states may also be covered by some predicate $C_i$, with a given maximum tolerable rate $f$. Therefore the abstraction may over-approximate the boundaries of an unsafe concrete region – thus including also safe concrete states – but will not misclassify unsafe regions. The abstraction is refined after each online learning phase to more accurately represent the unsafe regions.

**Abstract Model Construction and Minimisation.** After mapping each point in the concrete state space to an abstract state $C_i$, a finite surrogate MDP model $M_a$ can be constructed from the paths explored during online learning phases by estimating for each abstract state the applicable actions and the resulting transition probabilities to other abstract states. As a result of the safety invariant of the state abstraction process, $M_a$ may conservatively overestimate the probability of reaching an unsafe state, since some safe concrete states may be abstracted into unsafe abstract states.

In general, the surrogate state space may become intractably large, particularly for the purpose of counterexample generation. To mitigate this issue, we merge adjacent states, i.e., states connected by at least one transition/one edge, with similar safety behaviours to a single abstract state. The intuition behind this restriction is akin to spatial locality, in that states closer to an unsafe state are

more likely to lead to unsafe outcomes in future online phases. To define such a merging strategy, we define $\epsilon$-simulation for adjacent states – i.e., abstract states sharing at least one hyperplane – adapting the general notion of $\epsilon-$simulation (e.g., (Desharnais et al., 2008)):

**Definition 3.2** *Adjacent $\epsilon$-simulation: Let $S_D$ be the partition of $S$ induced by the equivalence relation $s \sim s'$ iff $L(s) = L(s')$. Then, for a given $\epsilon \in [0, 1]$, two adjacent states $s$ and $s'$ are $\epsilon$-similar if $S_D(s) = S_D(s')$ and $\forall s_D \in S_D \setminus S_D(s).\ |P(s, a, s_D) - P(s', a, s_D)| \leq \epsilon$.*

The $\epsilon$-simulation in Def. 3.2 allows the definition of a hierarchical merging scheme. Let level $l_0$ contain the initial abstract states. $\epsilon$-similar adjacent states from level $l_i$ are merged into a single state at level $l_{i+1}$, until no further merge is possible. Intuitively, since only adjacent abstract states can be merged, for a given $\epsilon$, the final minimised abstraction is likely to be finer in the neighbourhood of states labelled as unsafe (or as goal), where a small number of steps and corresponding traversal of adjacent states is sufficient to discriminate the outcome of the trajectory being explored; coarser when more steps are required to decide on the outcome.

**Counterexample-Guided Simulation.** For a safety requirement $P_{\leq \lambda}\ [F\ \texttt{unsafe}]$, we generate counterexamples of the safety specification based on the abstract, minimised model $M_A$ using the MILP introduced in Def. 2.2. All state-action pairs that may directly incur violating behaviours would be included in at least one of the multiple counterexamples generated from the abstract model, provided enough diverse counterexamples are generated.

Each generated counterexample submodel $M_{cex} = < S_{cex}, s_0, A_{cex}, P_{cex}, R_{cex}, L >$ is used as an abstract simulation environment aimed at penalising the selection of actions that are likely to lead to reaching an unsafe state. To this goal, the agent performs offline Q-Learning within $M_{cex}$ with a penalty-reward function $R_{cex}$ assigning a negative reward $-1$ upon reaching an unsafe state and $0$ when reaching other states – notice that, being $M_{cex}$ a sub-model of the abstraction $M_A$, some states may not have outgoing transitions since such transitions would not contribute to the violation of the safety requirement, as discussed in Sec. 2.

### 3.2 Simulation-based Learning with Counterexample Guidance

Algorithm 1 summarises the main steps of our simulation-based learning with counterexample guidance. We initially assume no knowledge about the environment is given to the agent; both the abstract model $M_A$ and the set of explored paths $D$ are empty (line 1). If prior knowledge is available, either in the form of an initial abstraction or of previously explored paths, $M_A$ and $D$ can be initialised accordingly.

The `onlineQLearning` procedure (line 3) lets the agent operate in the concrete environment similar to a standard Q-Learning process for $\sharp$ epochs, but with the following variation. At every step, if $M_A$ contains an abstraction for the current concrete state, with probability $\beta \in (0, 1)$ the agent selects the next action among those available from the corresponding abstract state, proportionally to their respective Q-value, while with probability $1 - \beta$ it selects a random action among all those available in the concrete space. Initially, the abstract model is empty, thus the agent will necessarily explore actions not represented in the abstract model. As the learning process progresses, the probability of exploring unknown actions will be bounded by $1 - \beta$. This allows potentially exploring new regions of the concrete space and refining the abstraction, while limiting within each online phase the intrinsic risk of exploring unknown regions. The online learning process can terminate earlier if the number of visits to unsafe states exceeds $\lfloor \lambda \cdot epochs \rfloor$.

The sequence of explored concrete paths, including the labelled concrete states visited, the action selected, and the collected rewards, is returned in $D_{online}$ and added to the experience dataset $D$ that will be used to refine the current abstraction $M_A$. The number of times unsafe concrete states are visited ($\sharp$ unsafe) is also returned and used to decide whether to trigger an offline learning phase to reduce the risk of reaching such discovered unsafe states in the future (line 6).

The offline learning phase starts with refining the abstraction $M_A$ (line 7), which is then used to generate a set of $N$ counterexamples, each used to train the agent offline until no significant changes to the Q-values are detected. It is in general possible to generate all diverse counterexamples from $M_A$ as described in Sec. 2. However, for large models, generating all of them may be prohibitively expensive and a finite number of counterexamples may be preferable.

Offline learning work on minimal submodels of the approximate, abstract environment each generated from a counterexample, therefore enclosing sufficient trajectories with enough probability mass to violate the safety requirements. Whenever the agent reaches an unsafe state during simulation, a negative reward gets backpropagated, ultimately penalizing the actions more likely to eventually

lead to a safety violation. No positive reward is given for reaching a goal during offline learning, positive rewards are instead collected during the online phase. In counterexample generation, there is a trade-off between covering the simulations most relevant to the current policy of the agent and producing a variety of diverse simulation environments. We pursue the first goal by weighting the objective function of the optimization problem in Def 2.2 with the current Q-values, thus preferring counterexamples closer to the current policy of the agent. We pursue the second goal (multiple counterexamples) by imposing a random blocking clause that excludes previously explored counterexamples during the same offline learning campaign in Sec. 2.

After each offline learning phase, the agent resumes online learning until convergence is achieved or a new offline phase is triggered by the occurrence of enough unsafe events. Different offline phases optimise the generation of counterexamples based on the Q-values updated from both online exploration and previous offline learning, thus increasing the likelihood of generating a different set of counterexamples that can reduce the exploration risk for the more likely action to be selected at that time.

---

**Algorithm 1:** Simulation-based Learning with Counterexample Guidance

1   $M_A, D \leftarrow \emptyset$ ;
2   **while** *not done* **do**
3     $D_{online}, \sharp\text{unsafe} \leftarrow \text{onlineQLearning}(M_A, \sharp epochs)$ ;    // online learning
4     $D \leftarrow D \cup D_{online}$ ;                        // update the dataset
5     $p_{\text{unsafe}} \leftarrow \sharp\text{unsafe}/\sharp epochs$ ;           // estimate safety rate
6     **if** $p_{unsafe} \geq \lambda$ **then**
       //simulation-based offline learning
7       $M_A \leftarrow \text{refine}(M_A, D)$ ;           // update the abstract model
8       **for** $t' \in 0, 1, \ldots, N$ **do**
9         $M_{\text{cex}} \leftarrow M_{\text{cex}}(M_A, \lambda)$ ;    // generate new counterexample simulation
10        $s \leftarrow s_0$ ;
11        $done \leftarrow Flsalse$ ;
12        **while** *not done* **do**
13          $a \leftarrow A_{\text{cex}}(s)$ ;        // sample action from the action space
14          $s', r, done \leftarrow M_{\text{cex}}(s, a)$ ;   // generate the destination state and reward with the simulation model
15          $Q(s, a) \leftarrow (1 - \alpha) * Q(s, a) + \alpha(r + \gamma * \max(Q(s')))$ ;
16          $s \leftarrow s'$ ;
17        **end**
18       **end**
19     **end**
20   **end**

---

**Theoretical discussion.** The switch between online and offline training reduces a possible shift between the Q-value computed on the abstract states and the Q-value that would result in learning directly in the concrete space thanks to the iterative refinement of the abstraction. As already discussed, the set cover solution preserves the *safety invariant* of Def. 3.1, thus discovered concrete unsafe states will always map to abstract unsafe states. We further analyse the convergence of the abstraction to an arbitrarily accurate representation of the concrete space throughout the learning process by showing that the following claims hold:

1. The probability of exploring any safe or unexplored concrete states is always non-zero
2. Eventually, all concrete unsafe states will be abstracted into an abstract unsafe state
3. The misclassification error for unexplored concrete state regions mapped into abstract states, while preserving the safety invariant, can be bounded with a Probably Approximately Correct (PAC) assumption (Valiant, 2013)

Since we always allow with a controllable probability $\beta$ the selection of any action available in a concrete state, we only reduce the likelihood of selecting potentially risky action, but we do not strictly prevent their selection as in, e.g., shielding methods (Alshiekh et al., 2018). In turn, every reachable concrete state will eventually be explored with probability 1 (claim 1).

When a previously unknown concrete unsafe state is explored (claim 2), the next refinement of the abstraction will preserve the safety invariant by mapping it to an unsafe abstract state (cf. Sec. 3.1).

Concerning claim 3, the abstraction error of the set cover solution in unexplored space can be bounded with $u \in (0, 1]$ with a probability of at least $1 - \delta$ with PAC assumption (Blumer et al., 1989). If the Vapnik–Chervonenkis(VC) dimension and the dimension $d$ of a learning concept $L$ is finite, then $L$ has a abstraction error at most $u$ with a probability at least $1 - \delta$ if $L$ is consistent with a sample size $\max(\frac{4}{u} \log \frac{2}{\delta}, \frac{8VC}{u} \log \frac{13}{u})$. According to the proof in (Blumer et al., 1989), VC dimension of a set cover solution $C$ in the form of $C = \{\cup_{i=1}^{s} C_i, 0 < i \le s\}$ is less than $2ds \log(3s)$, where $s$ is the number of set in C. Therefore, the required sample size to bound the abstraction error $u$ for a set cover classification is $\max(\frac{4}{u} \log \frac{2}{\delta}, \frac{16ds \log(3s)}{u} \log \frac{13}{u})$. Because the probability of exploring any concrete state remains strictly positive, the abstraction will eventually converge to any prescribed $(u, \delta)$ accuracy with probability 1.

## 4   Evaluation

We empirically evaluate the performance of our method from the perspective of 1) the improvement in the exploration safety rate and 2) the impact on the cumulative reward achieved by the agent.

**Environments.**    We consider three environments: slippery FrozenLake8x8 from OpenAI Gym (Brockman et al., 2016), DiscreteGrid environment from the implementation of  (Hasanbeig et al., 2022), and HybridGrid environment, where we change the state space of DiscreteGrid from discrete to continuous with the same layout. In all the environments, the agent decides a direction of move between up, down, left, and right. We define the objective of the agent as finding a policy with maximum Q-value while avoiding unsafe behaviours with intolerably high probability during exploration. In the *FrozenLake8x8* specifically, the agent aims to find a walkable path in an 8x8 grid environment with slippery actions while avoiding entering states labelled with *H*. With the slippery setting, the agent will move in the intended direction with a probability of only 1/3 else will move in either perpendicular directions with equal probabilities of 1/3 respectively. In the *DiscreteGrid* environment (Hasanbeig et al., 2022) and *HybridGrid* environments, the agent aims to reach the states labelled with "goal1" and then states labelled with "goal2" in a fixed order while avoiding entering any states labelled with "unsafe" along the path, with $15\%$ uncertainty of moving to random directions. In *HybridGrid*, the distance covered in a move is also randomly sampled from a Gaussian $\mathcal{N}(2, 0.5)$ thus moving in the same direction from a state may reach different states.

**Surrogate Model and Safety Specification.**    We parameterise the abstraction process with maximum false classification rate $f = 0.05$ for the set cover problem setting, $\epsilon = 0.01$ for adjacent $\epsilon$-simulation, and a discount factor $\gamma = 0.9$ for the Q-value updates. The minimisation of the abstract models reduces the state space by around 99 times (from 1600 to 15 abstract states) in DiscreteGrid. A similar reduction is observed for HybridGrid, where the abstraction can closely represent the grid layout of the continuous space. The resulting counterexamples are comparably small, thus introducing low overhead for both the MILP solution of the offline learning phases. Although the state space Frozenlake8x8 is originally small, the minimisation process simplifies the number of state space by $21.9\%$ (from 64 to 50 abstract states).

We set the safety specification parameter $\lambda = 0.2$ in DiscreteGrid and HybridGrid, and $\lambda = 0.35$ in FrozenLake8x8. These values are chosen to account for the intrinsic uncertainty in the respective environments, where the direction of movement chosen by the agent materialises with a probability $1/3$ in FrozenLake, and $85\%$ in DiscreteGrid and HybridGrid. In HybridGrid, also the length covered by each move is sampled from $\mathcal{N}(2, 0.5)$, thus introducing further uncertainty.

**Baselines.** We compare the learning performance of our method to the standard Q-Learning as the baseline without any guidance and compare our method with (Hasanbeig et al., 2018; 2022) (QL-LCRL in the figures), using the same set of hyper-parameters from their implementation. QL-LCRL is included in the evaluation for the intuition of combining model checking with model-free RL to encourage safety during learning, albeit they use model checking to shield them from unsafe actions rather than using counterexamples and offline learning.

**Experimental Results** Fig A.1 shows the accumulated rewards and safety rates in the three environments. The cumulative rewards of QL, QL-LCRL and QL-CEX (our method) all converge to similar values, showing that the performance under guidance (QL-LCRL and QL-CEX) remains level comparable to the baseline QL method. We observe that QL-CEX has a better average safety rate in all these three environments, and converges to higher safety rates faster than the other methods. In DiscreteGrid and HybridGrid, the safety rate of online learning in our method exceeds the

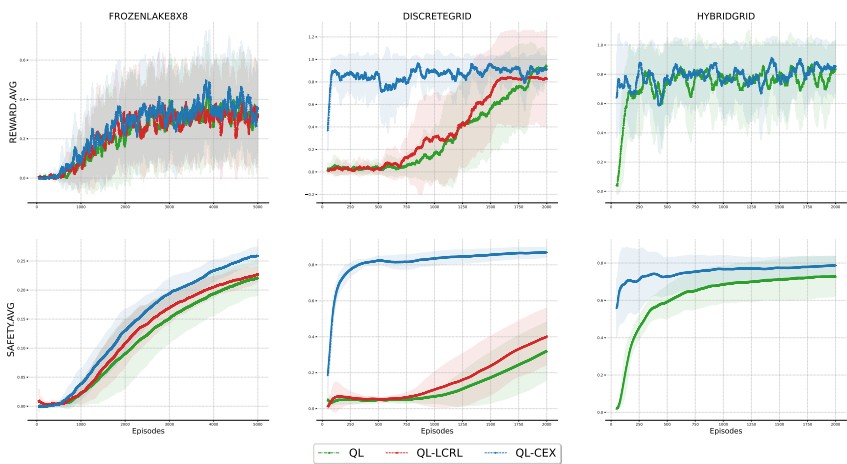

Figure 1: Average reward and safety rate with several experiments in QL, QL-LCRL (Hasanbeig et al., 2022) and QL-CEX (our method) in FrozenLake8x8, DiscreteGrid, and HybridGrid

threshold after the first iteration of offline learning. This is due to the rapid convergence of the abstraction thanks to the grid layout that can be accurately and efficiently represented and minimised. In FrozenLake8x8, more online data is required to form an accurate abstraction before the safety specification is satisfied. Although counterexample generation takes extra time in solving the MILP (averaged solve time for a single counterexample is less than 0.3 seconds for DiscreteGrid and HybridGrid and less than 2 seconds for FrozenLake8x8 on a Macbook Air with M1 CPU and 8Gb of memory, using Gurobi (Gurobi Optimization, LLC, 2022) as MILP solver), the offline training takes place within the counterexample-guided simulation model, which has a significantly smaller state space compared to the concrete model.

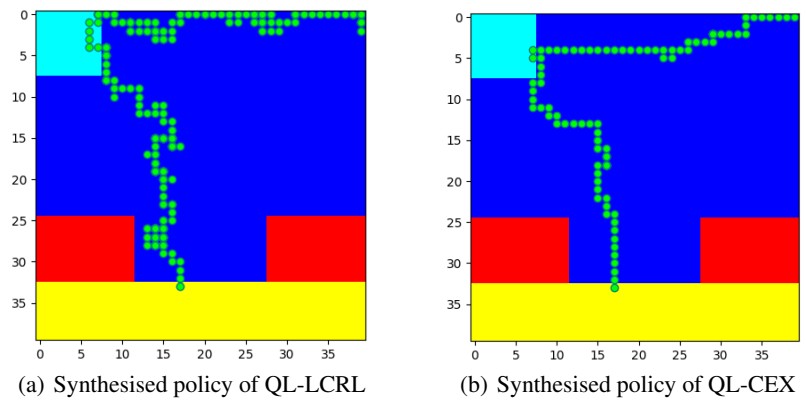

(a) Synthesised policy of QL-LCRL          (b) Synthesised policy of QL-CEX

Figure 2: Synthesised policies of QL-LCRL and QL-CEX in discrete DiscreteGrid

In the DiscreteGrid environment, the reduced abstract state space in QL-CEX significantly speeds up the learning process. We particularly show the synthesised policy of QL-LCRL and QL-CEX with the objective of finishing the two-steps task – reach goal 1 (cyan) and then goal 2 (yellow), avoiding the unsafe regions (red) – in order to compare the influence of the guidance methods. The QL-CEX policy leads to a smoother path towards the goal while steering further away from the unsafe regions. This is expected since shielding methods prevent the selection of actions deemed unsafe, while providing limited guidance for which alternative to select, while the offline training of QL-CEX leads the agent to adjust the Q-values of all explored actions to reduce the risk of eventually reaching an unsafe state.

Finally, for HybridGrid, the continuous concrete state space prevents the use of QL-LCRL (Hasanbeig et al., 2022), thus we compare only with QL as a baseline. Despite the uncertainty from both the 15% probability of moving in a random direction and the distance covered in such direction sampled from $\mathcal{N}(2, 0.5)$, thus potentially reaching a large set of different states after each action, QL-CEX rapidly converges to a safety rate around 70%. We remark that without accurate prior knowledge, it is not possible to entirely avoid visiting unsafe states during online exploration, since their location is not known before they are visited the first time. Our method aims at increasing the safety rate during online exploration, while it cannot totally prevent unsafe events from happening.

## 5 Related Work

**Safe Exploration.** (García & Fernández, 2015; Kim et al., 2020; Brunke et al., 2022) provide surveys and taxonomies of recent safe reinforcement learning methods with different emphases. Most existing safe exploration methods assume prior knowledge about the environment or the agent's dynamics (Moldovan & Abbeel, 2012; Fulton & Platzer, 2018), known safety constraints (Alshiekh et al., 2018; Jansen et al., 2018; Hasanbeig et al., 2018; Pham et al., 2018), or utilise expert guidance (Garcia & Fernández, 2012; Zhou & Li, 2018; Huang et al., 2018) to provide a guarantee for safety constraints satisfaction during exploration. A different class of methods (Sui et al., 2015; Wachi et al., 2018; Liu et al., 2020) utilises surrogate Gaussian process models to characterise the unknown dynamics and optimise the unknown function with *a prior* model structure. There are only a few methods (Dalal et al., 2018; Bharadhwaj et al., 2020) to tackle the unknown model structure by training a safety layer/critic used for filtering out probabilistic unsafe actions based on an offline dataset. While motivated by the same idea of safer exploration without prior knowledge, our method can be initialised with none or only a small amount of previously explored paths, meanwhile, show comparable or superior performance to other guidance-based algorithms (Hasanbeig et al., 2018) that rely on prior knowledge of the safety constraints.

**Offline RL.** Offline RL can be seen as a data-driven formulation of RL, where the agent collects transitions using the behaviour policy instead of interacting with the environment (Levine et al., 2020). The biggest challenge in offline RL is the bootstrapping error: the Q-value is evaluated with little or no prior knowledge and propagated through Bellman equation (Kumar et al., 2019). (Wu et al., 2019; Siegel et al., 2020) regularise the behaviour policy while optimising to address this issue. (Buckman et al., 2020; Kumar et al., 2020) alternatively update the Q values in more conservative ways to learn a lower bound of the Q function using uncertainty estimated with the sampled information. From the safe RL perspective, (Urpí et al., 2021) optimises a risk-averse criterion using data previously collected by a *safe* policy offline, and (Bharadhwaj et al., 2020) learn a conservative safety critic offline to tackle the problem of safe exploration with unknown MDPs. Our work instead introduces a pessimistic simulation model, maintaining conservative safety behaviours over the offline dataset and switching between online and offline learning to reduce the shift effect between simulation and concrete control systems.

## 6 Conclusion

We presented our investigation of a safer offline model-free reinforcement learning method using counterexample-guided simulation. We proposed an abstraction strategy to represent the knowledge acquired during online exploration in a succinct, finite MDP model that can consistently and accurately describe safety-relevant dynamics of the explored environment. Counterexample generation methods from probabilistic model checking are then adapted to synthesise small-scale simulation environment capturing scenarios in which the decisions of the agent may lead to the violation of safety requirements – the probability of reaching an unsafe state is higher than a tolerable threshold. The agent can then train offline within this simulated environment, using a reward scheme aimed at reducing the likelihood of selecting actions that may eventually lead to visiting again already discovered unsafe states. The knowledge refined during the offline training phases is then exploited to reduce the risk during subsequent online exploration, while newly explored paths feedback information to the next offline learning phase.

The alternation of online exploration and offline knowledge refinement and counterexample-guided learning can ultimately lead to higher safety rates during exploration, without significant reduction in the achieved cumulative reward, as demonstrated in our preliminary evaluation on problems from previous literature and the OpenAI Gym. While this work focused on improving Q-Learning, the basic framework is not specific to Q-Learning, and we plan to explore its impact on other learning algorithms in future work.

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

# A  Appendix

## A.1  Additional Evaluation Notes

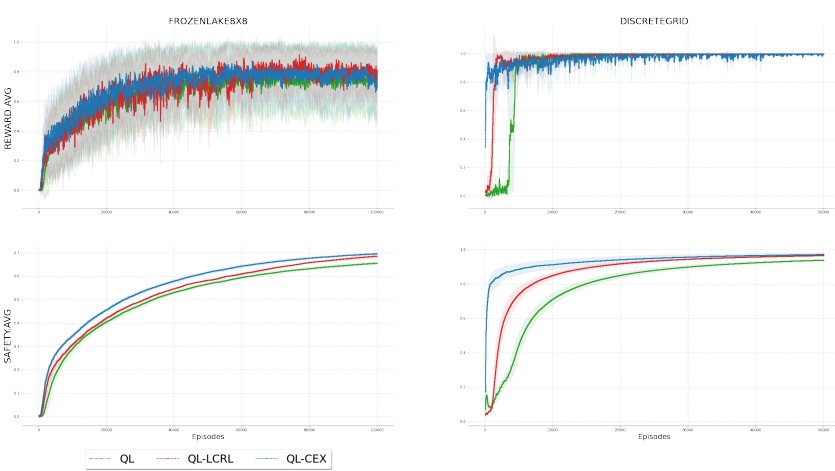

Figure 3: Average reward and safety rate with several experiments in QL, QL-LCRL (Hasanbeig et al., 2022) and QL-CEX (our method) in FrozenLake8x8 and DiscreteGrid with 100k online learning episodes

We compared the learning performances of our method and the other two baseline methods with three different environments in Sec. 4 with 5,000, 2,000, and 2,000 training episodes respectively. However, due to the high uncertainty the FrozenLake8x8 environment, none of the evaluated methods achieves the 65% safety rate threshold within 5,000 episodes while only QL-CEX (our method) achieves the required safety rate threshold (80%) in DiscreteGrid within 2,000 episodes. Fig. 3 shows the evaluation on FrozenLake8x8 and DiscreteGrid when 100,000/50,000 online learning episodes are allowed. It can be observed that both baseline methods and QL-CEX eventually converge to a policy that satisfy the safety requirement. However, QL-CEX has a generally faster safety rate improvement trend and already after a few thousand episodes, which reduces the number of unsafe events also throughout the subsequent online exploration phases. Notably, the offline learning phases are triggered if the preceding online phase violated the safety requirement. Hence, they should occur less frequently as exploration and abstraction consolidate.

## A.2  Implementation

All common hyper-parameters used with QL, QL-LCRL, and QL-CEX are the same. For QL-LCRL-specific additional hyper-parameters we use the same values as (Hasanbeig et al., 2022). The discount factor is $\gamma = 0.9$, confidence level of the cover solution $\sigma = 0.95$ (confidence) and $\epsilon = 0.01$ for adjacent $\epsilon$-simulation are used with same values for all environments. In the environment of slippery Frozenlake8x8, we set the learning rate $\alpha = 0.1$ and the initial value of a decaying probability of random exploration equal to 0.2, due to the high uncertainty in environment. In DiscreteGrid and HybridGrid, we set the learning rate $\alpha = 0.9$ and the initial value of a decaying probability of random exploration equal to 0.3.

To ease reproducibility and future comparisons, as well as reduce threats to the internal validity of our experiments due to bugs in our implementation, the code implementing QL-CEX is available at: `https://anonymous.4open.science/r/QL-CEX-70BD`.

