# OpenReview forum: "Safer Reinforcement Learning with Counterexample-guided Offline Training"
_ICLR.cc/2023/Conference — Submitted to ICLR 2023_

### Official Review · Reviewer_4QqX · 2022-10-19

**Confidence:** 3
**Correctness:** 4
**Technical Novelty And Significance:** 3
**Empirical Novelty And Significance:** 2
**Recommendation:** 3

**Clarity, Quality, Novelty And Reproducibility:**

The writing is unclear.

The paper quality is OK, and the algorithm is novel.
The results seem reproducible, although some implementation details seem sparse.

**Strength And Weaknesses:**

## Strengths

**************Method:**************

- The set cover formulation is very interesting and seems like a nice way to abstract states into sets of unsafe and safe states.
- The tie-in to VC dim/PAC learning analysis is nice, I wonder if the authors can show these bounds holding empirically?

************************Experiments:************************

- Experiments are solid, showing a higher safety rate and similar or sometimes higher rewards to sensible baselines.

## Weaknesses

**Clarity:**

- The background section could be written better.
    - “atomic propositions” are stated without definition in the first paragraph.
    - First sentence mentions a hybrid discrete-continuous state space, but then talks about $S$ being a finite state space. This is a bit confusing; the hybrid discrete-continuous state space can be introduced and then detailed later. I think it makes sense to talk about this only in Section 3.
    - The Probabilistic Model Checking section is confusing for someone (like me) who has not seen PCTL before. Rather than just listing everything and defining everything, after Eqs 2/3, it may make more sense to write an intuitive explanation of how PCTL is used for safety checking, define only the necessary terms, and then leave extra details to an appendix. Perhaps the authors could keep the definitions there, but explain how things are used. For example, they define the modal operator but do not mention how its connection to the RL safety problem. This applies for all operator definitions and makes this section very hard to read.
    - Again, in Definition 2.2, the main paper doesn’t need a huge, multi-line equation with a bunch of definitions. This should be left to the appendix, and it makes the paper harder to read. I’m not saying these equations are unnecessary, it’s just that the authors have presented them in a way that detracts from the clarity, and therefore quality, of this paper. Similarly, the explanation below details most of the equations and simply writes out what they are in words. This is unnecessary—-a higher-level intuitive explanation is better.
- Assumptions should be better stated beforehand: e.g. that the transition dynamics are assumed to be known.
- The word “simulation” may imply some kind of sim2real situation to certain groups of RL researchers. Perhaps instead of “simulation,” the authors can replace this with “offline learning phase.”
- Safety rate seems to be only defined in L5 of Alg 1, but talked about extensively in the experiments. The authors should properly define it in-text in the experiments section again.

**Experiments:**

- I know that this is not the point of the paper, but it would be nice to extend this algorithm to a continuous control task, the only difference being the underlying Q-learning algorithm would have to change to something like an actor-critic alg.
- Is there a possibility that the advantages over QL and QL-LCRL comes from the number of updates performed? DiscreteGrid8x8’s results make me a little suspicious that the results could be more similar if the other two baselines were matched in the ratio of Q-learning updates performed to environment steps.

****************************Minor details:****************************

- Typos:
    - “hybrid discrete-continues state space”
    - “Probabilistic model checking” background section: extra parenthesis after $\alpha$
- Questions:
    - A max false classification rate of 0.95 seems very high. Can the authors explain this choice?
- Some missing safety papers taking different approaches the authors could cite and compare: [https://arxiv.org/pdf/1702.01182.pdf](https://arxiv.org/pdf/1702.01182.pdf), [https://arxiv.org/abs/2008.06622](https://arxiv.org/abs/2008.06622),  [https://www.aaai.org/AAAI22Papers/AAAI-5810.MaY.pdf](https://www.aaai.org/AAAI22Papers/AAAI-5810.MaY.pdf)

**Summary Of The Paper:**

This paper proposes a method that extends Q-learning to train offline on counterexamples of unsafe states, generated through a safety specification combined with set covering algorithm.

**Summary Of The Review:**

I think this paper presents a nice method for safe RL in simple environments, however my main issue with this paper is that the writing is not clear enough, so I recommend a reject.

---

### Official Review · Reviewer_xUpK · 2022-10-24

**Confidence:** 4
**Correctness:** 2
**Technical Novelty And Significance:** 3
**Empirical Novelty And Significance:** 2
**Recommendation:** 3

**Clarity, Quality, Novelty And Reproducibility:**

**Clarity** This paper is overall easy to understand, but the theory part is extremely hard to follow. Given there is no formal theoretical argument or proofs, I don't think this paper is clearly written. I recommend the authors to rewrite this paper especially in terms of theory section.

**Quality** The proposed method is interesting and technically sound, but both theoretical and empirical results (or presentations) should be improved.

**Novelty**. I think the novelty of this paper is sufficient. As far as I know, the proposed approach is new and interesting.

**Reproducibility**. As for theory, I don't think sufficient information has been provided for understanding and reproducing the authors' theorem. As for experiments, the source code is not attached, and very small amount of information regarding experiments (e.g., hyper-parameters) has been written in the paper. Thus, I think the reproducibility of this paper is rather low.



**Strength And Weaknesses:**

### Strength
**Important problem to be solved**. I think this paper addresses an important problem for (safe) RL community.

**Interesting approach**. I think the proposed ideas are interesting and technically-sound. It is particularly nice to propose a method which does not require a prior knowledge and small amount of offline data.

### Weakness
**Theory**. There are theoretical discussions in page 6, but this part is quite unfriendly to the readers. The descriptions are informal, and there is no proof. I feel that most of the readers cannot catch up with the theoretical analyses in this paper. I guess this is due to the page limit, and I recommend the authors to
+ Clearly state each theorem with necessary assumptions. Also, provide proofs for theorems.
+ The authors can write proofs in the appendix if there is no space in the main paper.

**Experiment**. The benchmark problem is rather easy, so I think that the authors should have tested their method in continuous control problems as well as grid world. In this case, the baseline methods should be PPO-Lagrangian or CPO. The current experiments are conducted in limited problem settings; hence, I don't think that the authors' claims have been sufficiently supported empirically.

**Minor comments**
+ [Section 2]  A policy $\pi: s \rightarrow a$ $\Longrightarrow$ A policy $\pi: S \rightarrow A$
+ [Section 4] Ray & Amodei (2019) is the paper on Safety-Gym, so it is not appropriate to cite this for Slippery FrozenLake8x8
+ [Section 5] apriori --> a priori



**Summary Of The Paper:**

This paper addresses safe reinforcement learning (RL) problems for safety-critical applications where failures during learning may incur high costs. The authors aims to deal with (partially) unknown environments and dynamics, where an agent must discover unsafe state-action pairs during exploration. In this paper, the authors propose an abstraction strategy to represent the knowledge
acquired during online exploration in a succinct, finite MDP model that can consistently and accurately describe safety-relevant dynamics of the explored environment. The authors exploit probabilistic counterexamples generation to synthesize minimal, partial simulation environments from the surrogate model where the agent can train offline to produce heuristic strategies to minimize the risk of visiting unsafe states during subsequent online exploration. The effectiveness of their proposed method is demonstrated in several benchmark tasks.

**Summary Of The Review:**

Though the problem setting is interesting and the proposed approach is interesting, I have several concerns especially on theoretical and empirical results. I strongly recommend the authors to rewrite this paper by 1) improving the theory part (i.e., clearly state theorems and provide formal proofs) and 2) improving experiment part (i.e., add experiments on continuous control with strong baselines and provide source code or sufficient information for reproducibility).

---

### Official Review · Reviewer_QNmG · 2022-10-24

**Confidence:** 3
**Correctness:** 2
**Technical Novelty And Significance:** 2
**Empirical Novelty And Significance:** 1
**Recommendation:** 3

**Clarity, Quality, Novelty And Reproducibility:**

The writing quality of the paper can be improved, as the paper is a little bit difficult to understand.

- Is the $M_A$ in Algorithm 1 the same as $M_a$ in Section 3.2?
- Algorithm 1 L.15: $max \to \max$, and also mismatched parenthesis.
- What is $w$ in Eq 14? Is it fixed or learned or solved?
- Page 5: "lets the agent operates in the concrete environment" -> "lets the agent operate in the concrete environment"
- Abstract: "on a selection of problems form literature" -> "on a selection of problems from literature"
- Some details can be deferred to the appendix. For example, the syntax of PCTL and Definition 2.2.
- Definition 3.2: What does $\forall s_D \in S_D \backslash S_D(s)$ mean?
- It seems that QL-LCRL serves as an important baseline. Maybe describe how it works in a few sentences.
- I don't get how the set covering problem is solved.

Novelty: The high-level idea of constructing minimal dynamics to improve local safety seems novel to me, but I don't like the first step in QL-CEX, which is discretizing the state space.

Reproducibility: The authors provide a few hyperparameters but as the method is pretty complex and no code release is promised, I think reproducibility can be an issue.

**Strength And Weaknesses:**

My major concern with the paper is that it proposes an overcomplicated method, conducts experiments on toy examples and lacks rigid theoretical analysis.

- Method:
  - I'd say the QL-CEX is very sophisticated: It involves state abstraction, building and minimizing a dynamics model and finding counterexamples (i.e., sub-dynamics where the current policy visits unsafe states with probability greater than $\lambda$).
  - The algorithm invokes the `onlineQLearning` procedure, but the standard Q Learning only works for discrete state/action space. How does it work for the HybridGrid environment? Furthermore, is the $Q$ in Algorithm 1, L.15 the same $Q$ used in L.3? Note that the real environment has a different reward function than $R_{cex}$ and even the state space is different.
  - I didn't understand why PCTL is used for probabilistic model checking. As far as I can see, the proposed method discretizes state space (in the sense that every state is mapped to a discrete abstract state) and also learns a model. So I think it might be possible to reuse the existing work on safety RL in a tabular case.

- Experiments:
  - The experiments are simple and can only serve as proof of concept. I don't see an obvious way to scale up QL-CEX. As far as I can see, QL-CEX converts a continuous state space to a discrete state space, which means that it essentially can't deal with a continuous state space.
  - The paper uses QL-LCRL as a baseline, the experiments show that it's only slightly better than the standard Q Learning algorithm in DiscreteGrid and even worse than it. Any other good baselines?
  - In FrozenLake8x8, all of the three algorithms don't satisfy the safety constraint: the one that performs best only achieves ~25% safety rate, which is far from the goal (65% if I understand it correctly).
  - This paper lacks ablation studies. For example, how well do geometric abstraction and model construction work?
- Theory:
  - I don't expect a theoretical analysis of state abstraction, which is not the central topic of this paper. I expect a theoretical analysis of other properties of QL-CEX, e.g., convergence, safety guarantee, and sample efficiency.

**Summary Of The Paper:**

This paper studies the problem of safe reinforcement learning. It proposes a model-based reinforcement learning algorithm to address the problem, by abstracting the states and constructing a minimal environment to improve the local safety of the learned policy. The small environment is constructed by finding "counterexamples" using probabilistic model checking. Experiments on three toy environments from related literation and OpenAI Gym are conducted to show the efficacy of the proposed algorithm.

**Summary Of The Review:**

As stated above, I'm concerned about the complexity of the proposed method,  the insufficiency of the experiments and the lack of rigid theoretical analysis. I don't recommend acceptance in its current form.

---

### Official Review · Reviewer_noqT · 2022-10-25

**Confidence:** 2
**Correctness:** 3
**Technical Novelty And Significance:** 3
**Empirical Novelty And Significance:** 2
**Recommendation:** 3

**Clarity, Quality, Novelty And Reproducibility:**

- Writing is clear for most parts, however, for audiences unfamiliar with formal verification methods literature, it might be a bit hard to follow the paper.
- The contributions seem novel.


**Strength And Weaknesses:**

Strengths of the paper:

- The paper is well-written and succinctly introduces the notion of model-based safety.
- The counterfactual construction of safety specification seems elegant, though I have to admit I am not very well-versed in the literature on formal verification and model-based safe RL.
- Empirical results look promising.

Some questions to the authors:

- Why did authors restrict the baselines to Hasanbeig et. al 2018; 2022? Why aren't baselines from offline RL, for ex: RISK-AVERSE OFFLINE REINFORCEMENT LEARNING, ICLR'21 used?

**Summary Of The Paper:**

The authors propose a safe reinforcement learning algorithm for the setting where the environment dynamics are unknown or partially known. A hybrid offline-online learning method is proposed where some online exploration is done to construct a notion of safety in that environment. The authors use a probabilistic model checker to produce counterfactual sub-models, which are potentially unsafe states. Finally, simulation-based learning is used to train the agent offline to avoid reaching unsafe states.



**Summary Of The Review:**

- Authors propose a model-based safe RL method, with an online exploration component to construct a counterfactual sub-model to learn about unsafe actions and states via probabilistic model checking.
- Experiments results on model-based safe RL baseline demonstrate the efficacy of the proposed method.

--- Review update ---

Thank you authors for all the efforts. I have read the reviews by other reviewers, and I agree with the following broad limitations of the paper:

- Complex method, without satisfactory experimental results.
- Lack of rigid theoretical analysis.
- As pointed out by Reviewer xUpK, the theory introduced in the paper is very hard for the readers to follow, so the writing needs a lot of improvements.

Based on the lack of strong experimental results, and writing, I am changing my scores of the paper. I recommend authors change the writing and experimental section based on the reviews.

---

> ### Comment · Reviewer_noqT · 2022-11-24
> **Review update**
>
> Thank you authors for all the efforts. I have read the reviews by other reviewers, and I agree with the following broad limitations of the paper:
>
> - Complex method, without satisfactory experimental results.
> - Lack of rigid theoretical analysis.
> - As pointed out by Reviewer xUpK, the theory introduced in the paper is very hard for the readers to follow, so the writing needs a lot of improvements.
>
> In summary, as per the other reviews, based on the lack of strong experimental results, and unclear writing, I am changing my scores for the paper. I recommend authors change the writing and experimental section based on the feedback from the reviews.

---

### Decision · Program_Chairs · 2023-01-20

**Decision:**

Reject

**Justification For Why Not Higher Score:**

Lack of proper theoretical or empirical support for the method proposed in the paper.

**Justification For Why Not Lower Score:**

N/A

**Metareview: Summary, Strengths And Weaknesses:**

Although the reviewers believe the paper is addressing an important problem in safe RL, they are seriously concerned about the lack of proper theoretical analysis and experimental evaluation. The theory part of the paper is not written well, the descriptions are informal. The authors should properly state their theorems with all the necessary assumptions and then prove them either in the paper or in the appendix. Such as informal description of the theoretical findings is not convincing. The experiments are simple and their results are not convincing either. With all the above, I do not think the paper is ready for publication and requires more work and improvement.